# FBSVP: Video Prediction Based on Foreground-Background Separation

## Abstract

Video prediction is the process of learning necessary information from historical frames to predict future video frames. How to focus and efficiently learn features from historical frames is a critical step in this process. For any sequence of video frames, the background changes little or remains almost constant, while the foreground changes significantly and is the main focus of our video prediction learning. However, current known video prediction learning methods do not consider how to utilize the different characteristics of the foreground and background to further improve prediction accuracy. To fully leverage the different characteristics of the foreground and background and enhance prediction accuracy, we propose a Foreground-Background Separation Video Prediction (FBSVP) model in this paper. Through the foreground and background separation module, historical video frames are separated into foreground and background frames. In the video prediction module, the foreground and background frames are predicted and learned separately. First, the features of historical frames are fused into the current frame through a historical attention fusion module using an attention mechanism. Then, the complementary temporal and spatial features are fused through a spatio-temporal fusion module. Finally, the learned foreground and background features are fused in the foreground and background fusion module to predict the final video frame. Experimental results show that our proposed FBSVP model achieves the best performance on popular video prediction datasets, demonstrating its significant competitiveness in this field.

## 1 Introduction

Video can be seen as a special type of temporal data that is well-suited for modeling using Recurrent Neural Networks (RNNs). The work by Ranzato et al. (2014) first utilized RNNs to model the spatiotemporal dynamics of videos in an unsupervised manner, which inspired a series of subsequent studies Finn et al. (2016); Srivastava et al. (2015); Oliu et al. (2018); Zhang et al. (2019). However, RNN-based approaches primarily focus on capturing temporal features of videos while overlooking spatial information. To address this limitation, Convolutional Neural Networks (CNNs) were introduced Shi et al. (2015) to complement the RNNs, resulting in the widely adopted hybrid architecture of convolutional and recurrent layers in most video prediction models Shi et al. (2017); Wang et al. (2017; 2018b; 2019); Guen & Thome (2020); Ballas et al. (2015). This hybrid architecture allows models to leverage the ability of convolutional units to model spatial relationships and the potential of recurrent units to capture temporal dependencies. Although popular in the literature, these classical video prediction architectures still have two main limitations. Firstly, in dense prediction tasks like video prediction, models need to have a sufficiently large receptive field to capture rich contextual information. Previous works attempted to enlarge the receptive field of prediction units through 3D convolutions Wang et al. (2018a); Yu et al. (2020), but the receptive field is primarily determined by the kernel size of the integrated convolutional operators. Increasing the receptive field would significantly increase the model's memory consumption and computational cost. Secondly, existing video prediction models struggle to effectively fuse captured spatial and temporal features to enhance prediction accuracy.

Many current approaches simplify the training process by independently modeling these two features Villegas et al. (2017); Denton et al. (2017), only performing simple fusion when generating predicted frames. In reality, spatial and temporal features are complementary, and fully integrating both features during training is crucial to better understand the patterns of video variations and improve the model's perception ability.

To address the above issues, we propose a video prediction model based on Foreground-Background Separation (FBSVP). Due to the differences in the characteristics of the video frame foreground and background, we separate the foreground and background of the video frames and then predict them separately. This allows for more effective video prediction tailored to their respective characteristics and enables more focused and efficient learning of video frame motion patterns. It avoids the interference caused by different feature changes, which can lead to a decrease in prediction performance. Since separate prediction for the foreground and background reduces the complexity of the prediction, it helps to lower the difficulty of prediction, naturally improving the accuracy. Finally, the more accurately predicted foreground and background features are effectively fused to produce the final predicted video frame. Experimental results show that the proposed FBSVP outperforms other state-of-the-art methods in major video prediction tasks.

## 2 Related Work

### 2.1 Video Prediction

The latest research progress in video prediction provides some useful insights into how to predict future visual frames based on historical observations. In this section, we will discuss recent advancements in video prediction methods. Ranzato et al. (2014) utilized recurrent neural networks (RNN) to model videos based on a language model. Srivastava et al. (2015) proposed FC-LSTM, an improved variant of RNN with long short-term memory (LSTM) that enhances the model's ability to capture temporal dependencies in videos. Shi et al. (2015) introduced ConvLSTM, which replaces the fully connected layers in FC-LSTM with convolutional layers to improve perception of visual data and save parameters. Similarly, Ballas et al. (2015) employed convolutional layers with gated recurrent units (GRU) for video prediction. However, Wang et al. (2017) argued that both temporal and spatial information should be equally considered and proposed ConvLSTMs (ST-LSTM) with spatial modules to model the spatial representation of each frame. They further introduced Casual LSTM Wang et al. (2018a) to increase the temporal depth of the model and Gradient Highway Unit to alleviate gradient propagation issues in deep prediction models. Guen & Thome (2020) introduced PhyCell, which separates physical dynamics from unknown factors to predict more reliable motion. Additionally,Wu et al. (2021) proposed Motion-GRU to independently model transient changes and motion trends for more satisfactory predictions.

Despite the significant achievements of the aforementioned methods, the models still have relatively narrow receptive fields, making it challenging to capture rich contextual information and improve the perception ability of video features.

### 2.2 Foreground-Background Separation

Foreground-background separation methods have been designed and proposed in many computer vision tasks (Cristani et al., 2010; Garcia-Garcia et al., 2020; Zhao et al., 2023; Ding et al., 2022; Yang et al., 2020; Liu et al., 2023). Shao et al. (2022) proposed a foreground-background separation (FBS) X-ray contraband detection framework, using an attention module to make the detection framework more focused on the foreground. The proposed framework can separate contraband items as the foreground from other irrelevant items using only available bounding box labels and accurately identify contraband items in severely occluded and overlapped X-ray images. This demonstrates that separating the foreground and background and focusing more on the foreground can effectively improve model performance. Zhang et al. (2022) proposed a foreground-background separation mutual generative adversarial network (FSM-GAN) framework for video anomaly event detection, which can identify the spatio-temporal features of the foreground under background conditions and

achieve satisfactory results even on large-scale datasets. Yang et al. (2021) believe that the foreground and background should be treated equally and proposed a collaborative video object segmentation method through a multi-scale foreground-background integration (CFBI+) approach, improving the results of video object segmentation. This indicates that the relationship between the foreground and background is inseparable and complementary.Besides the aforementioned papers, there are also other related excellent papers(An et al., 2023; Li et al., 2023; Moayeri et al., 2022).

Inspired by the excellent performance of foreground-background separation methods in various applications, this paper proposes a video prediction model based on foreground-background separation (FBSVP) to enhance video prediction performance.

## 3 Method

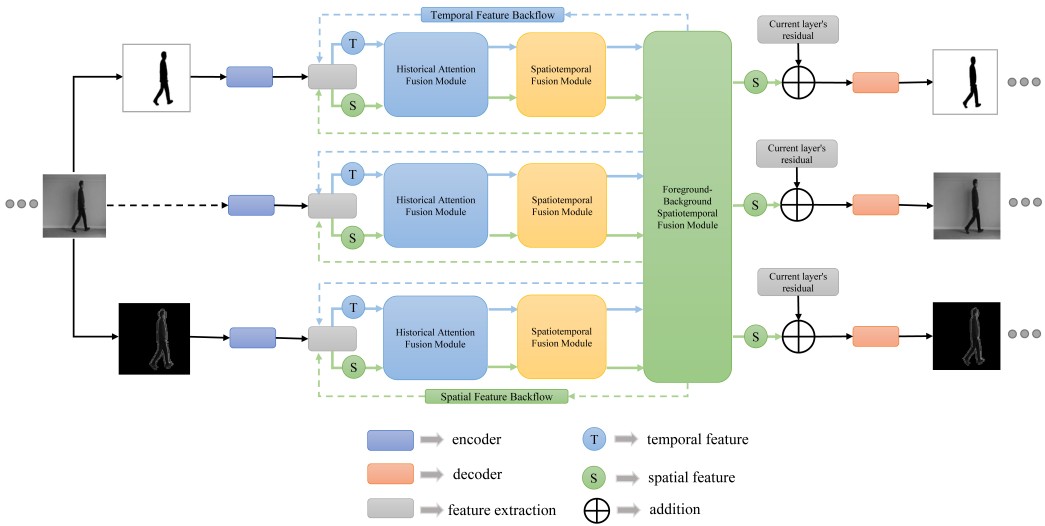

Figure 1: The structure of the single-layer stacked FBSVP.

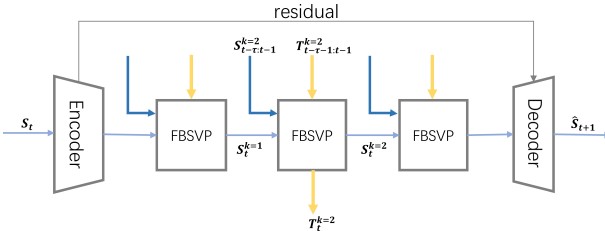

Figure 2: The structure of the predictive network with stacked FBSVPs.

### 3.1 Foreground-background video frame extraction

Currently, the methods for extracting the foreground and background mainly come from the open-source toolkit provided by OpenCV, which includes seven different algorithms: KNN (Zivkovic & Van Der Heijden, 2006) (K-nearest neighbors) based on the K-nearest neighbors algorithm, MOG (KaewTraKulPong & Bowden, 2002)/MOG2 (Zivkovic, 2004)(Mixture of Gaussians) based on the mixture of Gaussians algorithm, CNT (Counting) based on pixel counting algorithm, GMG (Godbehere et al., 2012) based on pixel color feature algorithm,

LSBP (Guo et al., 2016)(Local SVD Binary Pattern) based on local SVD binary pattern algorithm, and GSOC (Google Summer of Code) algorithm similar to LSBP. Among these, KNN and MOG2 have the best practical application results. This paper chooses MOG2 because this algorithm can more flexibly adjust parameters according to the scene to adapt to different situations.

## 3.2 Encoder

As shown on the left side of Figure 1, FBSVP uses a 2D convolutional encoder to process the input video frames, encoding the original video frames, foreground video frames, and background video frames separately. The output of each layer is connected to the decoder through residual connections, providing the decoder with the necessary residual features.

## 3.3 Foreground-background Separation Prediction

In this section, we will provide a detailed description of the structural details of FBSVP, as shown in the middle part of Figure 1. The Foreground-background Separation Prediction Module consists of three fusion modules: Single-layer Historical Attention Fusion Module, Single-layer Spatiotemporal Fusion Module, and Foreground-background Spatiotemporal Fusion Module. Typically, to enhance the model's expressive and perceptual capabilities, multiple FBSVPs are stacked, as depicted in Figure 2. It is important to note that at time step $t$ in the $k$th layer, FBSVP has a total of three inputs: spatial features $S_t^{k-1}$ from the $k-1$th layer, accumulated spatial features $S_{t-\tau:t-1}^k$ from the $k$th layer over the previous $\tau$ time steps, and accumulated temporal features $T_{t-\tau-1:t-1}^k$ from the $k$th layer over the previous $\tau + 1$ time steps.

Here is a unified convention for the symbol notation: $S$ represents spatial features, $T$ represents temporal features, the superscript $s$ denotes parameters related to spatial feature calculations, the superscript $t$ denotes parameters related to temporal feature calculations, the superscript $k$ denotes the $k$-th layer, and the subscript $t$ denotes the time instant. In the superscript or subscript, $(i)$ with $i = 1, 2, 3...$ is used to distinguish the different states of the same algorithmic symbol at different stages of the model.

### 3.3.1 Single-layer Historical Attention Fusion Module

Spatial feature information and temporal feature information complement each other. To fully capture both temporal and spatial features, we introduce an attention mechanism. The goal is to assist prediction units in giving different attention to different historical temporal and spatial features. Since temporal and spatial features influence each other, the attention given to temporal features helps capture a portion of spatial features, while the attention given to spatial features helps capture a portion of temporal features. This way, spatial and temporal features can learn from each other, enhancing the model's perceptual capabilities.

Based on the above analysis, the attention score $M_j^s$ for temporal features can be represented as follows, $j$ represents the $j$-th attention score. Among them $i = 1, \ldots, \tau$, $j = 1, \ldots, \tau$:

$$S_t' = W_{(1)}^s * S_t^{k-1} \;, m_i^s = SUM\left(S_{t-i}^k \odot S_t'\right) \;, M_j^s = \frac{e^{m_j^s}}{\sum_{i=1}^{\tau} e^{m_i^s}} \tag{1}$$

Similarly, the attention score $M_j^t$ for spatial features can be represented as follows:

$$T_{t-1}' = W_{(1)}^t * T_{t-1}^k \;, m_i^t = SUM\left(T_{t-i-1}^k \odot T_{t-1}'\right) \;, M_j^t = \frac{e^{m_j^t}}{\sum_{i=1}^{\tau} e^{m_i^t}} \tag{2}$$

Where $SUM$, $\odot$, and $*$ represent summation, Hadamard product, and convolution operations, respectively. By using the computed attention scores, we obtain a portion of spatial feature information $S_{att\_part}$ and temporal feature information $T_{att\_part}$.

$$T_{att\_part} = \sum_{j=1}^{\tau} M_j^s \cdot T_{t-j-1}^k, S_{att\_part} = \sum_{j=1}^{\tau} M_j^t \cdot S_{t-j}^k \tag{3}$$

we integrate the attention-based spatial feature $S_{att\_part}$ and the attention-based temporal feature information $T_{att\_part}$ into the corresponding spatial and temporal features, respectively.

$$F_{(1)}^t = sigmoid\left(T_{t-1}'\right), \quad F_{(1)}^s = sigmoid\left(S_t'\right) \tag{4}$$

$$T_t^{(1)} = F_{(1)}^t \odot T_{t-1}^k + \left(1 - F_{(1)}^t\right) \odot T_{att\_part} \tag{5}$$

$$S_t^{(1)} = F_{(1)}^s \odot S_t^{k-1} + \left(1 - F_{(1)}^s\right) \odot S_{att\_part} \tag{6}$$

### 3.3.2 Single-layer Spatiotemporal Fusion Module

Temporal and spatial features are inseparable components of video features, reflecting the changing patterns of video features from two different perspectives. They complement each other, and the fusion of spatio-temporal features promotes mutual perception and learning between the two. This further enhances the model's perception capabilities. To optimize the integration of temporal and spatial features, we apply a convolutional transformation to the previously fused temporal feature $T_t^{(1)}$ and spatial feature $S_t^{(1)}$ obtained from the previous module.

$$T_t'' = W_{(2)}^t * T_t^{(1)}, \quad S_t'' = W_{(2)}^s * S_t^{(1)} \tag{7}$$

Subsequently, we merge the temporal and spatial features.

$$F_{(2)}^t = sigmoid\left(T_t''\right), F_{(2)}^s = sigmoid\left(S_t''\right) \tag{8}$$

$$T_t^{(2)} = F_{(2)}^t \odot T_t'' + \left(1 - F_{(2)}^t\right) \odot S_t'', S_t^{(2)} = F_{(2)}^s \odot S_t'' + \left(1 - F_{(2)}^s\right) \odot T_t'' \tag{9}$$

$$S_t^{(2)} = F_{(2)}^s \odot S_t'' + \left(1 - F_{(2)}^s\right) \odot T_t'' \tag{10}$$

It is important to note that the main difference between historical attention fusion and spatiotemporal fusion is that historical attention fusion requires calculating attention scores based on the temporal and spatial features of the current video frame and several past video frames, based on their interrelatedness. This guides the model to learn key features from the past video frames with different weights. In contrast, spatiotemporal feature fusion is relatively straightforward, where a portion of the temporal features and a portion of the spatial features are computed and summed together to ensure the mutual integration of features.

### 3.3.3 Foreground-background Spatiotemporal Fusion Module

In the first half of the process, to reduce the mutual interference of features learned by the model, the foreground and background features are trained separately, which helps to focus more on learning the motion patterns of video frames and enhances the model's prediction capabilities. To predict the actual video frames, it is necessary to fuse the learned foreground and background features. The foreground and background features are two important and inseparable characteristics of a video frame, influencing and complementing each other. The foreground features can indirectly reflect the characteristics of the background features, and similarly, the background features can indirectly reflect the characteristics of the foreground features.

Therefore, for models that adopt separate training for the foreground and background, it is crucial to thoroughly fuse the learned foreground and background features. Since we have designed a model that learns three features simultaneously, it is important to consider learning the fusion of these features while learning the foreground and background features. This approach enables better prediction of the actual video frames.

In this segment, to enhance clarity of expression, the following conventions are made: $\{foreground|merge|background\}$ represents a choice between foreground, merge, or background levels but for a complete formula, only one can be selected - either all foreground, all merge, or all background. To simplify the expression of formulas, we use $f$ to represent foreground, $m$ to represent merge, and $b$ to represent background. Thus,

$\{foreground|merge|background\}$ is simplified to $\{f|m|b\}$. In the following text, $f$, $m$, or $b$ appearing in superscripts or subscripts will represent foreground, medium, or background, respectively. The subscript $t\_\{f|m|b\}\_level$ represents an abstract feature at one of the foreground, medium, or background levels at time instant $t$, superscript $s\_\{f|m|b\}\_level$ or $t\_\{f|m|b\}\_level$ represents computational parameters related to spatial or temporal feature computation at one of the foreground, merge, or background levels, Both $\{S|T\}$ and $\{s|t\}$ represent either selecting spatial features or temporal features, but for a complete formula or diagram, either all $S$ and $s$ are chosen or all $T$ and $t$ are chosen.

Prior to fusion, a convolutional transformation is applied to the abstract features at each layer.

$$T^{(3)}_{t\_\{f|m|b\}\_level} = W^{t\_\{f|m|b\}\_level}_{(3)} * T^{(2)}_{t\_\{f|m|b\}\_level} \tag{11}$$

$$S^{(3)}_{t\_\{f|m|b\}\_level} = W^{s\_\{f|m|b\}\_level}_{(3)} * S^{(2)}_{t\_\{f|m|b\}\_level} \tag{12}$$

$$F^{\{s|t\}\_\{f|m|b\}\_level}_{(3)} = sigmoid\left(\{S|T\}^{(3)}_{t\_\{f|m|b\}\_level}\right) \tag{13}$$

The spatiotemporal features of the foreground are integrated separately with the fused features and the spatiotemporal features of the background, as shown in Figure 3: The fused

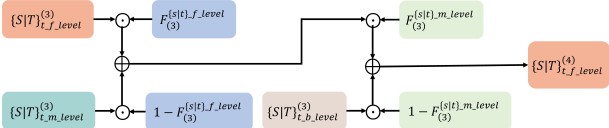

Figure 3: The algorithmic diagram illustrating the fusion of foreground features with the fused and background features shows the fusion process across these three levels, following the direction of the arrows.

spatiotemporal features are fused separately with the foreground and background spatiotemporal features, as shown in Figure 4. The spatiotemporal features of the background are

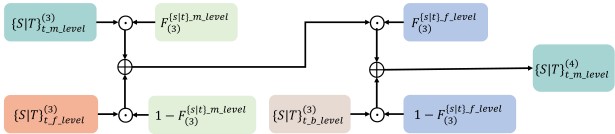

Figure 4: The diagram illustrating the fusion of fused features with the foreground and background features shows the fusion process across these three levels, following the direction of the arrows.

fused separately with the fused and foreground spatiotemporal features, as shown in Figure 5. In this case, the extracted spatiotemporal features go through three fusion modules: the

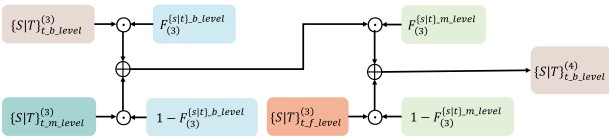

Figure 5: The diagram illustrating the fusion of background features with the fused and foreground features shows the fusion process across these three levels, following the direction of the arrows.

single-layer historical attention feature fusion module, the single-layer spatiotemporal fusion module, and the multi-layer foreground-background spatiotemporal fusion module. It can

be observed that this model fully perceives and integrates the spatiotemporal features of the video. The foreground and background are trained and predicted separately, significantly reducing interference in feature learning. This allows for a more focused approach to feature learning, resulting in more accurate predictions and demonstrating the model's powerful perception and prediction capabilities.

### 3.4 Decoder

As shown on the right side of Figure 1, the decoder architecture corresponds to a mirrored version of the convolutional encoder. It encodes the predicted original video frames, foreground video frames, and background video frames separately. The features in the residual connections are fused with the decoded feature maps through channel concatenation. Due to the extensive feature fusion, the most recent spatial feature maps already incorporate temporal feature maps. To maintain consistency with the encoder, the decoder ignores predicted temporal feature maps that are absent in the encoder's input. Ultimately, the decoder generates the next predicted video frame for the original, foreground, and background video frames separately. These three predicted frames serve as the basis for preparing the prediction of the next frame.

## 4 Experiments

### 4.1 Experimental Setups

In this section, extensive experiments will be conducted to evaluate the performance of the proposed model compared to state-of-the-art methods. We evaluate the proposed FBSVP on five different video datasets with varying levels of complexity, namely Moving MNIST (Srivastava et al., 2015), TrafficBJ (Zhang et al., 2017), KTH (Schuldt et al., 2004), KITTI (Geiger et al., 2013), Caltech Pedestrian (Dollár et al., 2009). Furthermore, all models are implemented using PyTorch and optimized using the Adam optimizer (Kingma & Ba, 2014) on a single Tesla P100 (16GB) GPU. Table 6 summarizes the more detailed experimental settings for the aforementioned tasks, In this context, Train and Test represent the number of input and predicted frames during training and testing, respectively. Layers indicate the number of stacked prediction units.

### 4.2 Video Prediction

#### 4.2.1 Moving MNIST

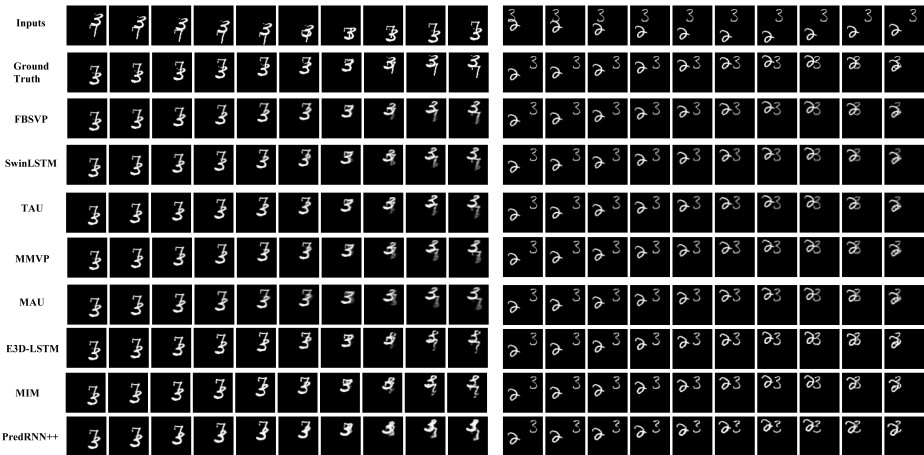

Figure 6: Predictions on the Moving MNIST dataset (10 frames → 10 frames) by different methods.

| Moving MNIST | | |
| --- | --- | --- |
| Method | SSIM/frame↑ | MSE/frame↓ |
| ConvLSTM(NeurIPS2015)(Shi et al., 2015) | 0.707 | 103.3 |
| FRNN(ECCV2018)(Oliu et al., 2018) | 0.819 | 68.4 |
| VPN(ICML2017)(Kalchbrenner et al., 2017) | 0.870 | 70.0 |
| PredRNN(NeurIPS2017)(Wang et al., 2017) | 0.869 | 56.8 |
| PredRNN++(NeurIPS2018)(Wang et al., 2018a) | 0.898 | 46.5 |
| MIM(CVPR2019)(Wang et al., 2019) | 0.910 | 44.2 |
| E3D-LSTM(ICLR2019)(Wang et al., 2018b) | 0.910 | 41.3 |
| Conv-TT-LSTM(NeurIPS2020)(Su et al., 2020) | 0.915 | 53.0 |
| MAU(NeurIPS2021)(Chang et al., 2021) | 0.937 | 27.6 |
| PhyDNet(ICLR2020)(Guen & Thome, 2020) | 0.947 | 24.4 |
| SimVP(CVPR2022)(Gao et al., 2022) | 0.948 | 23.8 |
| MMVP(CVPR2023)(Zhong et al., 2023) | 0.952 | 22.2 |
| SimVPv2(Tan et al., 2022) | 0.952 | 21.81 |
| TAU(CVPR2023)(Tan et al., 2023) | 0.957 | 19.8 |
| SwinLSTM(ICCV2023)(Tang et al., 2023b) | 0.962 | 17.7 |
| FBSVP w/o FBS | 0.958 | 18.9 |
| FBSVP w/ FBS | 0.963 | 16.2 |

Table 1: Quantitative results on the Moving MNIST dataset (10 frames → 10 frames) for different methods

| TrafficBJ | | | |
| --- | --- | --- | --- |
| Method | MSE × 100↓ | MAE↓ | SSIM↑ |
| ConvLSTM(NeurIPS2015)(Shi et al., 2015) | 48.5 | 17.7 | 0.978 |
| PredRNN(NeurIPS2017)(Wang et al., 2017) | 46.4 | 17.1 | 0.971 |
| PredRNN++(NeurIPS2018)(Wang et al., 2018a) | 44.8 | 16.9 | 0.977 |
| MIM(CVPR2019)(Wang et al., 2019) | 42.9 | 16.6 | 0.971 |
| E3D-LSTM(ICLR2019)(Wang et al., 2018b) | 43.2 | 16.9 | 0.979 |
| PhyDNet(ICLR2020)(Guen & Thome, 2020) | 41.9 | 16.2 | 0.982 |
| SimVP(CVPR2022)(Gao et al., 2022) | 41.4 | 16.2 | 0.982 |
| SimVPv2(Tan et al., 2022) | 34.8 | 15.6 | 0.984 |
| TAU(CVPR2023)(Tan et al., 2023) | 34.4 | 15.6 | 0.983 |
| FBSVP w/o FBS | 33.5 | 15.5 | 0.982 |
| FBSVP w/ FBS | 32.1 | 15.2 | 0.984 |

Table 2: Quantitative results of different methods on the TrafficBJ dataset(4 frames → 4 frames)

Figure 6 illustrates prediction examples from different methods, and compared to other methods, the proposed FBSVP (Foreground-Background Separation Video Prediction) achieves predictions with the best visual quality, significantly outperforming other methods. Particularly, it obtains notably better results in the last two time steps, indicating the superior expressive power of the proposed model. Additionally, Table 1 summarizes detailed quantitative results, where Mean Squared Error (MSE) and Structural Similarity Index (SSIM) are used to indicate the visual quality of the predictions. Lower MSE and higher SSIM scores suggest better visual quality. Compared to other existing methods, the proposed FBSVP achieves the best performance.

### 4.2.2 TrafficBJ

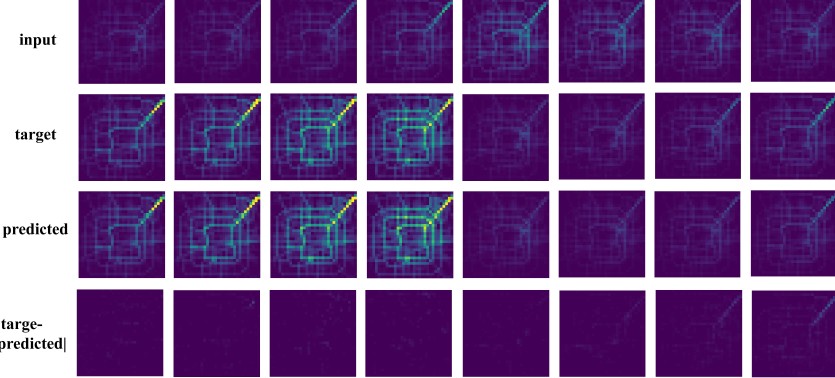

Figure 7: Qualitative visualization of the prediction results on the TrafficBJ dataset.

We present the quantitative results in Table 2 and qualitative results in Figure 7. Despite the significant differences between the given frames and the future frames, our model can still generate accurate and reliable frames. To make the comparisons more evident, we also visualize the differences between the actual frames and the predicted frames in the last row. Clearly, FBSVP exhibits the best performance among all the compared models, with the lowest intensity of differences in all predicted frames.

### 4.2.3 KTH

We used Peak Signal-to-Noise Ratio (PSNR) and Structural Similarity Index (SSIM) as evaluation metrics to measure the quality of frame prediction from a perceptual perspective. The quantitative results are shown in Table 3. It can be observed that FBSVP outperforms other methods in both PSNR and SSIM metrics. Furthermore, FBSVP even demonstrates

accurate prediction of future frames in long-range scenarios, such as 10 frames → 40 frames, showcasing its ability to predict future frames with flexible lengths.

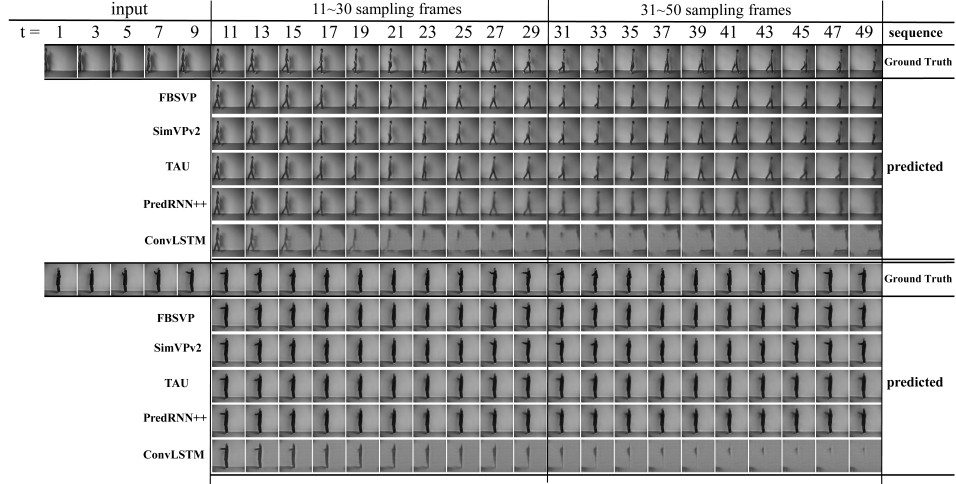

Figure 8: Prediction samples of KTH dataset, forecasting 40 future frames based on observing 10 frames.

| KTH | | | | |
|---|---|---|---|---|
| | KTH(10 → 20) | | KTH(10 → 40) | |
| Method | SSIM↑ | PSNR↑ | SSIM↑ | PSNR↑ |
| Mcnet(ICLR2017)(Villegas et al., 2017) | 0.804 | 25.95 | 0.730 | 23.89 |
| ConvLSTM(NeurIPS2015)(Shi et al., 2015) | 0.712 | 23.58 | 0.639 | 22.85 |
| DFN(NeurIPS2016)(Jia et al., 2016) | 0.794 | 27.26 | 0.652 | 23.01 |
| FRNN(ECCV2018)(Oliu et al., 2018) | 0.771 | 26.12 | 0.687 | 23.77 |
| PredRNN(NeurIPS2017)(Wang et al., 2017) | 0.839 | 27.55 | 0.703 | 24.16 |
| PredRNN++(NeurIPS2018)(Wang et al., 2018a) | 0.865 | 28.47 | 0.741 | 25.21 |
| E3D-LSTM(ICLR2019)(Wang et al., 2018b) | 0.879 | 29.31 | 0.810 | 27.24 |
| STMFANet(CVPR2020)(Jin et al., 2020) | 0.893 | 29.85 | 0.851 | 27.56 |
| SwinLSTM(CVPR2023)(Tang et al., 2023a) | 0.903 | 34.34 | 0.879 | 33.15 |
| SimVP(CVPR2022)(Gao et al., 2022) | 0.905 | 33.72 | 0.886 | 32.93 |
| MMVP(CVPR2023)(Zhong et al., 2023) | 0.906 | 27.54 | 0.888 | 26.35 |
| TAU(CVPR2023)(Tan et al., 2023) | 0.911 | 34.13 | 0.897 | 33.01 |
| SimVPv2(Tan et al., 2022) | 0.913 | 34.24 | 0.895 | 33.35 |
| FBSVP w/o FBS | 0.916 | 30.45 | 0.902 | 29.72 |
| FBSVP w/ FBS | 0.917 | 30.92 | 0.903 | 29.84 |

Table 3: Quantitative results of different methods on the KTH dataset(10 frames → 20 frames and 10 frames → 40 frames)

| Caltech Pedestrian | | | |
|---|---|---|---|
| Method | MSE($10^{-3}$)↓ | SSIM↑ | PSNR↑ |
| BeyondMSE(ICLR2016)(Mathieu et al., 2015) | 3.42 | 0.847 | - |
| MCnet(ICLR2017)(Villegas et al., 2017) | 2.50 | 0.879 | - |
| CtrlGen(ICLR2018)(Hao et al., 2018) | - | 0.900 | 26.5 |
| PredNet(ICLR2017)(Lotter et al., 2016) | 2.42 | 0.905 | 27.6 |
| ContextVP(ECCV2018)(Byeon et al., 2018) | 1.94 | 0.921 | 28.7 |
| E3D-LSTM(ICLR2019)(Wang et al., 2018b) | 2.12 | 0.914 | 28.1 |
| rCycleGan(CVPR2019)(Kwon & Park, 2019) | 1.61 | 0.919 | 29.2 |
| CrevNet(ICLR2020)(Yu et al., 2020) | 1.55 | 0.925 | 29.3 |
| STMFANet(CVPR2020)(Jin et al., 2020) | 1.59 | 0.927 | 29.1 |
| MAU(NeurIPS2021)(Chang et al., 2021) | 1.34 | 0.939 | 29.4 |
| SimVP(CVPR2022)(Gao et al., 2022) | 1.56 | 0.940 | 33.1 |
| TAU(CVPR2023)(Tan et al., 2023) | 1.52 | 0.946 | 33.7 |
| SimVPv2(Tan et al., 2022) | 1.48 | 0.949 | 33.2 |
| FBSVP w/o FBS | 1.21 | 0.952 | 31.2 |
| FBSVP w/ FBS | 1.17 | 0.953 | 32.1 |

Table 4: Quantitative results of different methods on the Caltech Pedestrian dataset (10 frames → 1 frame)

In Figure 8, we present prediction samples from different methods. Compared to other methods, our proposed FBSVP demonstrates more accurate prediction of human actions in long-term forecasting, with the best visual quality and a clear superiority over other methods. This indicates that the proposed model possesses strong capabilities in long-term prediction.

### 4.2.4 KITTI and Caltech Pedestrian

The quantitative results presented in Table 4 indicate that our proposed method achieves state-of-the-art performance in the generalization evaluation task, as measured by the MSE, SSIM, and PSNR metrics. In Figure 9, we present qualitative visualization results, where the last column showcases the visual differences between actual frames and predicted frames. It can be observed that our model accurately predicts changes in lighting conditions and lane markings, with minimal disparities between the predicted and actual frames. This demonstrates the strong predictive capabilities of FBSVP.

|  input  |  target  |  predicted  |  |target- predicted|  |
|:---:|:---:|:---:|:---:|

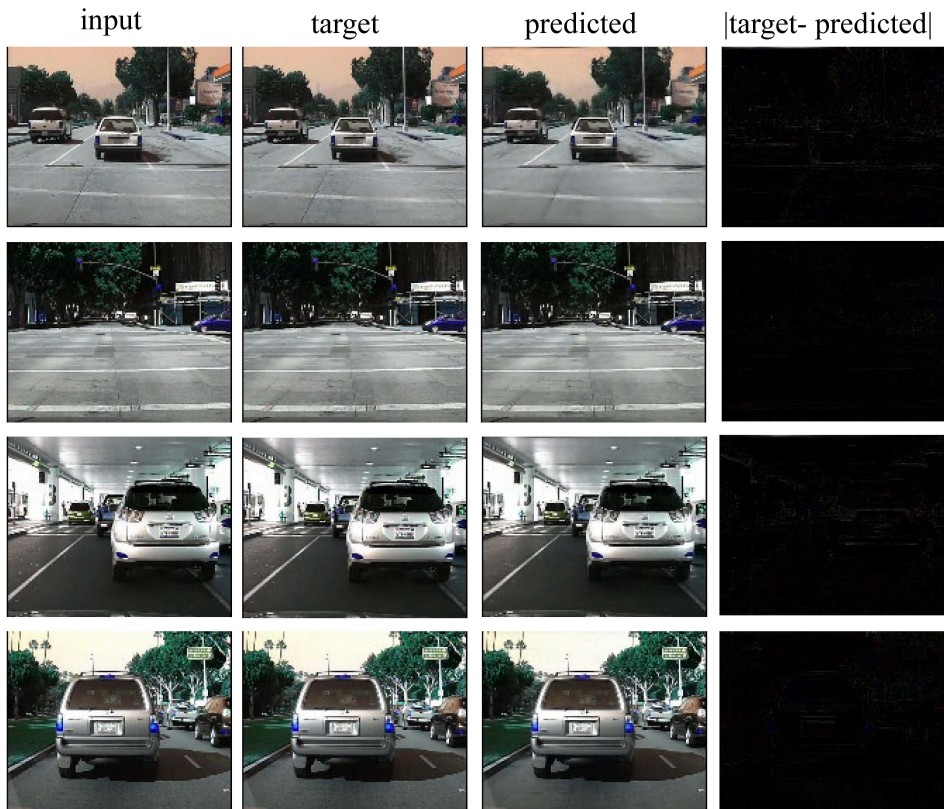

Figure 9: Qualitative visualization of prediction results on the Caltech Pedestrian dataset.

## 5 Ablation Study

### 5.1 FBSVP model architecture

We investigated the importance of different module design choices in the FBSVP model. Specifically, we studied the relevance of the temporal and spatial hierarchical structures and the impact of different fusion methods used in the prediction unit on the prediction results. For our ablation study, we focused on the Moving MNIST dataset. The results of our ablation study are listed in Table 5, with the best-performing results highlighted in bold and the second-best results underlined. As shown in Table 5, $s\_att\_fuse$ represents

Table 5: Ablation experiment results

| | FBSVP Modules | | | | | | Results | | | |
|---|---|---|---|---|---|---|---|---|---|---|
| rownum | s_att_fuse | s_t_fuse | t_att_fuse | b_s_t_fuse | f_s_t_fuse | f_b_s_t_fuse | MSE↓ | SSIM↑ | PSNR↑ | LPIPS↓ |
| 1 | ✓ | - | - | - | - | - | 31.1 | 0.929 | 22.27 | 6.52 |
| 2 | ✓ | ✓ | - | - | - | - | 27.8 | 0.938 | 22.77 | 5.32 |
| 3 | ✓ | ✓ | ✓ | - | - | - | 23.8 | 0.947 | 23.63 | 4.61 |
| 4 | ✓ | ✓ | ✓ | ✓ | - | - | 20.8 | 0.953 | 23.78 | 4.27 |
| 5 | ✓ | ✓ | ✓ | - | ✓ | - | 17.7 | 0.959 | 24.19 | 3.39 |
| 6 | ✓ | ✓ | ✓ | - | - | ✓ | 16.2 | 0.963 | 24.78 | 3.11 |

the historical spatial attention fusion module, $t\_att\_fuse$ represents the historical temporal attention fusion module, $s\_t\_fuse$ represents the single-layer spatio-temporal fusion module, $b\_s\_t\_fuse$ represents feature fusion of only the background, $f\_s\_t\_fuse$ represents feature fusion of only the foreground, $f\_b\_s\_t\_fuse$ represents feature fusion of both the foreground and the background. Additionally, ✓ represents the model selecting the corresponding module, - represents the model not selecting the corresponding module.

From the comparison of the last three rows in the table, it is easy to discover that foreground features contribute more to improving prediction accuracy than background features. It is necessary to pay more attention to foreground features. At the same time, foreground and background features complement each other and are inseparable. Combining both together can better enhance the performance of the prediction model.

## 5.2 Generalization capability

We selected relatively easy-to-modify video prediction models: ConvLSTM(Shi et al., 2015), PredRNN++(Wang et al., 2018a), MIM(Wang et al., 2019), E3D-LSTM(Wang et al., 2018b), and MAU(Chang et al., 2021). We modified these models according to the FB-SVP model approach, allowing them to predict separately using foreground and background separation and then fuse the results to obtain the final prediction. All experiments were conducted on the Moving MNIST dataset, and we used MSE and SSIM as comparison metrics. The experimental results are shown in Figures 10 and 11. In these figures, "RAW" represents the training results of the original models, and "FBSVP" represents the training results of the modified models. From the experimental results, it can be seen that the prediction performance of all modified models has been significantly improved, indicating that the proposed FBSVP model can serve as a general method to enhance the accuracy of video prediction.

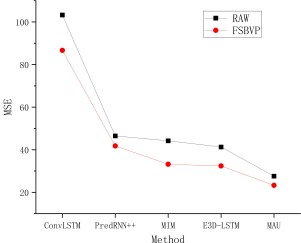 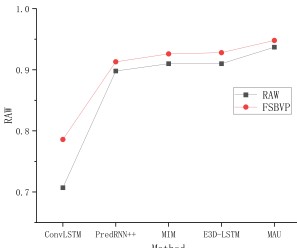

Figure 10: Experimental results of MSE metrics for different models

Figure 11: Experimental results of SSIM metrics for different models

## 6 Conclusion

In this paper, we propose a video prediction model based on foreground-background separation (FBSVP). By training the foreground and background features separately, FBSVP can effectively avoid the mutual interference that occurs during the joint learning of different features, which often leads to a decrease in prediction performance. This approach also allows the model to focus more on the relatively important foreground features, enabling it to better learn the motion characteristics of video frames. To fully learn and fuse the features of video frames, we designed three different fusion modules: the historical attention fusion module, the spatio-temporal fusion module, and the foreground-background spatio-temporal fusion module. The latter module re-fuses the previously separately trained foreground and background features to predict the actual video frames. The proposed model was evaluated on major video prediction tasks, and the experimental results demonstrate that our FBSVP model achieves the best performance on popular video prediction datasets, showcasing its significant competitiveness in the field.

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

## A    Preliminaries

The spatiotemporal prediction learning problem is defined as follows. Given a video sequence $G^{t,T} = \left\{ g^{\mathrm{i}} \right\}_{t-T+1}^{t}$ at time $t$ with the past $T$ frames, the goal is to predict the subsequent $T'$ frames $P^{t+1,T'} = \left\{ g^{\mathrm{i}} \right\}_{t+1}^{t+1+T'}$ from time $t + 1$, where $G$ is the past ground-truth frames, $P$ is the predicted future frames and $g^{\mathrm{i}} \in \mathbb{R}^{C \times H \times W}$ is typically an image with channels $C$, height $H$, and width $W$. In practice, video sequences are often represented as tensors, i.e., $G^{t,T} \in \mathbb{R}^{T \times C \times H \times W}$ and $P^{t+1,T'} \in \mathbb{R}^{T' \times C \times H \times W}$.

The model with learnable parameters $\boldsymbol{\Theta}$ learns the mapping $\mathcal{F}_{\boldsymbol{\Theta}} : G^{t,T} \mapsto P^{t+1,T'}$ by exploring spatial and temporal dependencies. In this paper, the mapping $\mathcal{F}_{\boldsymbol{\Theta}}$ is a neural network model that is trained to minimize the difference between predicted future frames and actual future frames. The optimal parameters are denoted as $\boldsymbol{\Theta}^*$.

$$\boldsymbol{\Theta}^* = \arg \min_{\boldsymbol{\Theta}} \mathcal{L} \left( \mathcal{F}_{\boldsymbol{\Theta}} \left( G^{t,T} \right), P^{t+1,T'} \right)$$

Where $\mathcal{L}$ is the loss function used to evaluate such differences.

## B    MORE DETAILS ABOUT DATASETS

### B.1    Moving MNIST

The Moving MNIST dataset is a standard dataset for video prediction. Each sequence in the dataset consists of 20 consecutive frames with a resolution of 64×64. Each sequence shows how two random digits move at a constant speed and bounce within the 64x64 frames. The handwritten digits are randomly sampled from the MNIST dataset (LeCun, 1998). By assigning different initial positions and velocities to each digit, an infinite number of sequences can be generated, allowing us to accurately evaluate the performance of each model without worrying about data scarcity. In the default setting, the models are trained to predict the future 10 frames after observing the first 10 frames in the sequence. Although

the movement in Moving MNIST may seem simple at first glance, generating consistent future frames in long-term prediction tasks can be quite challenging, as the digits may frequently bounce or occlude each other. We use a Moving MNIST generation script to generate Moving MNIST sequences from the standard MNIST training set. The models are tested on the official Moving MNIST test set.

### B.2 TrafficBJ

Traffic flow prediction is of great significance for traffic management and public safety, while being highly challenging due to various complex factors. We consider traffic flow prediction as a fundamental problem in spatio-temporal forecasting. Previous methods for traffic flow prediction have suffered from low prediction quality due to the complex dependencies on road networks and nonlinear dynamics.

Traffic flow data is collected from the chaotic real-world environment. They do not change uniformly over time, and there is a strong temporal dependency between the traffic conditions at adjacent timestamps. We use the TrafficBJ dataset (Zhang et al., 2017) to evaluate the traffic prediction capability of our proposed model. TrafficBJ contains trajectory data of Beijing collected from taxi GPS, where each frame is a $32 \times 32 \times 2$ image grid with two channels, namely inflow and outflow as defined in Zhang et al. (2017). Following previous works Wang et al. (2019); Guen & Thome (2020), we normalize the data to [0,1] using min-max normalization. The training model predicts the subsequent 4 frames by observing the previous 4 frames.

### B.3 KTH

The KTH Action Dataset (Schuldt et al., 2004) consists of six types of human actions (walking, jogging, running, boxing, waving, and clapping), performed multiple times by 25 subjects in four different scenarios: outdoors, outdoors with scale variation, outdoors with different clothing, and indoors. All video clips were recorded with a static camera at a frame rate of 25fps on a homogeneous background, with an average duration of four seconds. To ensure comparability, we followed the experimental settings in Wang et al. (2017; 2018b); Villegas et al. (2017) by resizing the video frames to $128 \times 128$ pixels. The dataset was divided into a training set (persons 1-16) and a test set (persons 17-25), with all models trained on the training set using all six action categories. The models were trained to predict the next 20 or 40 frames based on observations from the previous 10 frames. The challenge of this human motion prediction task lies not only in its flexible prediction length but also in the complex dynamics involving the randomness of human intention.

### B.4 KITTI and Caltech Pedestrian

Generalization ability is one of the fundamental challenges in artificial intelligence technology, particularly in unsupervised environments, which is a core research focus in machine learning. To evaluate the generalization ability of the proposed FBSVP model, we assess its prediction results across different datasets through spatiotemporal forecasting learning. KITTI (Geiger et al., 2013) is one of the most popular datasets for mobile robotics and autonomous driving. It consists of several hours of traffic scenes recorded using high-resolution RGB images. Caltech Pedestrian (Dollár et al., 2009) is a driving dataset focused on pedestrian detection, containing approximately 10 hours of $640 \times 480$ 30 FPS videos captured from vehicles driving in urban environments. Following the experimental setup in Yu et al. (2020); Lotter et al. (2016), the proposed model is trained on the KITTI dataset and tested on the Caltech Pedestrian dataset. The frame rate of the Caltech Pedestrian dataset is adjusted to match KITTI (10 FPS). All frames in both datasets are center-cropped and resized to $128 \times 160$. Furthermore, the proposed model is trained to predict the next frame based on the previous 10 frames as input. During testing, the prediction time horizon is extended to 10 frames.

## C    MORE DETAILS ABOUT EXPERIMENTAL SETTINGS

Table 6: Experimental settings for video prediction tasks on different datasets

| Experimental Settings | | | | |
|---|---|---|---|---|
| Dataset | Resolution | Train | Test | Layers |
| Moving MNIST | 64×64×1 | 10 → 10 | 10 → 10 | 4 |
| TrafficBJ | 32×32×1 | 4 → 4 | 4 → 4 | 2 |
| KTH | 128×128×1 | 10 → 20 | 10 → 20 | 4 |
| | | 10 → 40 | 10 → 40 | |
| KITTI | 128×160×3 | 10 → 10 | - | 8 |
| Caltech Pedestrian | 128×160×3 | - | 10 → 10 | 8 |

