# OpenReview forum: "FBSVP: Video Prediction Based on Foreground-Background Separation"
_ICLR.cc/2025/Conference — Submitted to ICLR 2025_

### Official Review · Reviewer_EHRB · 2024-10-29

**Soundness:** 2
**Presentation:** 2
**Contribution:** 2
**Rating:** 3
**Confidence:** 4

**Summary:**

This paper proposes a video prediction method that trains by separating foreground and background. The method uses MOG2 to separate foreground from background and then predicts future frames by fusing historical frame information into the current frame. The authors validated the algorithm on datasets such as Moving MNIST and TrafficBJ.

**Strengths:**

1. The authors achieved some performance improvement on multiple datasets, such as TrafficBJ and KTH.
2. The authors conducted both qualitative and quantitative analyses of the proposed method's performance, producing some reasonably good visual results.

**Weaknesses:**

1. The proposed idea of training by separating foreground and background has certain limitations. It is based on the assumption that backgrounds change infrequently while foregrounds change frequently (see Lines 014–016), which holds mainly for simple scenes, such as those in datasets like Moving MNIST and KTH used in this paper. In complex scenes, however, both the background and foreground can undergo significant changes, and foreground objects can vary greatly in size and spatial position within the frame (e.g., appearing larger when close, smaller when far). For scenes with complex backgrounds, foreground-background separation itself becomes challenging, and performing video prediction on separately processed foreground and background could be difficult. It would be helpful if the authors could analyze the applicability of this method in complex scenes or test on more challenging datasets such as UCF101 and Kinetics400.
2. Video prediction can be considered a subtask of mainstream video generation, where the goal is to condition on the first few frames of a video to generate future content. The methods compared by the authors are all based on supervised video prediction, lacking comparisons with mainstream video generation methods like Stable Video Diffusion, OpenSora, and VideoCrafter. Current mainstream video generation methods tend to generalize better in handling complex scenes, with a broader applicability than the supervised methods commonly used in video prediction. It would be beneficial if the authors could compare their approach with mainstream video generation models on datasets representing more general and complex scenes.
3. Miscellaneous: Figure 1 is too small, and the text is difficult to read. It would be better to use a larger image with adjusted font size for clarity.

**Questions:**

1. The core selling point of this paper is the prediction of video content by separately processing foreground and background. If the foreground-background separation algorithm is inaccurate, how does this affect the final result? Have the authors conducted any quantitative or qualitative analysis on the performance of the foreground-background segmentation algorithm and its impact on subsequent video prediction? For example, the authors could perform an ablation study using different foreground-background separation algorithms of varying accuracy.
2. I noticed that the captions for Table 1 and Table 2 indicate that video frame prediction is performed by conditioning on 10 frames to predict 10 frames (Table 1) or conditioning on 10 frames to predict 1 frame (Table 2). Have the authors tested results under other settings, such as conditioning on *n* frames to predict *m* frames, with various combinations of *n* and *m*? Testing with different combinations might provide a more comprehensive reflection of the method's effectiveness and robustness in video prediction.

---

> ### Author Response · Authors · 2024-11-27
> **Provide additional explanations and rebuttals to the reviewers' comments**
>
> For Weaknesses：
>
> 1、The proposed idea of training by separating foreground and background has certain limitations........
>
> ==>answer:I completely agree with your point. This is indeed a concern I had during the experimental process. However, when training on the KITTI dataset and testing on the Caltech Pedestrian dataset, the proposed model still performs excellently in real-world, complex scenarios. Therefore, the proposed model is applicable to complex scenarios. It would be even better to further validate the model on other complex datasets such as UCF101 and Kinetics400, as it would help further confirm the model's adaptability to complex environments.
>
> 2、Video prediction can be considered a subtask of mainstream video generation, where the goal is to condition on the first few frames of a video to generate future content.........
>
> ==>answer: Your point makes a lot of sense. Indeed, the comparison with recent mainstream video generation methods still leans towards more traditional supervised video prediction. In future work, we should not limit ourselves to a relatively narrow scope. It's important to broaden our perspective, and considering the integration of video generation methods could also be a promising direction. Thank you very much for your suggestion
>
> 3、Miscellaneous: Figure 1 is too small, and the text is difficult to read. It would be better to use a larger image with adjusted font size for clarity.
>
> ==>answer: Thank you very much for your suggestion. Previously, in order to reduce the space occupied by the images, I resized them, not realizing that it would affect the reading experience. I have now made the revisions as per your suggestion.
>
> For Questions:
>
> 1、The core selling point of this paper is the prediction of video content by separately processing foreground and background.......
>
> ==>answer: The foreground-background separation algorithm is inaccurate, which can affect the final result. The current model uses a foreground-background separation algorithm that has been derived through many experiments. Ablation studies have actually been conducted, and they should indeed be explicitly presented to help readers better understand. Thank you very much for your suggestion.
>
> 2、I noticed that the captions for Table 1 and Table 2 indicate that video frame prediction is performed by conditioning on 10 frames to predict 10 frames.......
>
> ==>answer: I completely agree with your idea. This would certainly provide a more comprehensive reflection of the effectiveness and robustness of the proposed method in video prediction. However, since I mainly referenced other related video prediction papers, such as [1] CVPR 2023 'Temporal Attention Unit: Towards Efficient Spatiotemporal Predictive Learning' and [2] CVPR 2022 'SimVP: Simpler yet Better Video Prediction,' among a series of other papers, this is how they handled it. I followed their typical approach for performance-related testing.

---

### Official Review · Reviewer_LpEN · 2024-10-30

**Soundness:** 2
**Presentation:** 3
**Contribution:** 2
**Rating:** 5
**Confidence:** 4

**Summary:**

A proper Foreground-Background Separation Video Prediction (FBSVP) model, has been proposed
in this paper, to enhance prediction accuracy in video prediction.

The features of historical (assumed to be past) frames are fused into the current frame through a historical attention fusion module using an attention mechanism.

Then, the complementary temporal and spatial features are fused through a spatio-temporal fusion module. Finally, the learned foreground and background features are fused in the
F/G-B/G fusion module to predict the final frame.
Experimental results shows that the proposed FBSVP model achieves the
best performance on few benchmark video prediction datasets, demonstrating its
power.

**Strengths:**

Good set of results on benchmark datasets; beats few SoA processes
proper analytics given for the process.

Ablation studies also provided.

Method clearly explained.

**Weaknesses:**

State clearly the difference between the operations:
historical attention fusion
vs
Spatio-temporal fusion.

Difference between the terms information and feature (T & S) should be clarified - unless used interchangeably (specify so then).

What about the use of any probabilistic model in prediction, future may be uncertain (although GT data may provide unique answer) ?

Fig. 1- text inside processing blocks barely legible

Rightmost column of Fig. 7 - is mostly dark - not sure, what the authors want to show to readers.

Only for Kitti D/S the number of test frames is larger than training 10 -> 40, well not actually so, as I am pointing out to.

The real challenge lies in predicting:
(i)  the number of predicted frames is large enough (longer duration), compared to history/past;
(ii) HR frames from LR ones - did not find any such results

(i) is partly addressed, although better to show results on say:
train: 10 --> 20
and then
test 10--> 40 or 80/120 - what happens then.?





F/G-B/G separation is well-studied topic with vast literature, which is often difficult to cover in a Conf. paper.
However,
a few relevant papers which have not been cited, or performance not compared, are given below:

Y. Zhao, D. Luo, F. Wang, H. Gao, M. Ye and C. Zhu, "End-To-End Compression for Surveillance Video With Unsupervised Foreground-Background Separation," in IEEE Transactions on Broadcasting, vol. 69, no. 4, pp. 966-978, Dec. 2023, doi: 10.1109/TBC.2023.3280039.

Motion-aware Contrastive Video Representation Learning via Foreground-background Merging; Shuangrui Ding et. al; CVPR – 2022

Collaborative Video Object Segmentation by Foreground-Background Integration; Zongxin Yang, Yunchao Wei, Yi Yang; ECCV 2020; https://www.ecva.net/papers/eccv_2020/papers_ECCV/html/3385_ECCV_2020_paper.php
(although a later T-PAMI paper, has been cited).

Revisiting Foreground and Background Separation in Weakly-supervised Temporal Action Localization: A Clustering-based Approach; Qinying Liu, Zilei Wang, Shenghai Rong, Junjie Li, Yixin Zhang; Proceedings of the IEEE/CVF International Conference on Computer Vision (ICCV), 2023, pp. 10433-10443

ZBS: Zero-Shot Background Subtraction via Instance-Level Background Modeling and Foreground Selection; Yongqi An, Xu Zhao, Tao Yu, Haiyun Guo, Chaoyang Zhao, Ming Tang, Jinqiao Wang; Proceedings of the IEEE/CVF Conference on Computer Vision and Pattern Recognition (CVPR), 2023, pp. 6355-6364

Motion-Aware Contrastive Video Representation Learning via Foreground-Background Merging; Shuangrui Ding, Maomao Li, Tianyu Yang, Rui Qian, Haohang Xu, Qingyi Chen, Jue Wang, Hongkai Xiong; Proceedings of the IEEE/CVF Conference on Computer Vision and Pattern Recognition (CVPR), 2022, pp. 9716-9726

Mitigating and Evaluating Static Bias of Action Representations in the Background and the Foreground;  Haoxin Li, Yuan Liu, Hanwang Zhang, Boyang Li; Proceedings of the IEEE/CVF International Conference on Computer Vision (ICCV), 2023, pp. 19911-23.

A Comprehensive Study of Image Classification Model Sensitivity to Foregrounds, Backgrounds, and Visual Attributes;  Mazda Moayeri, Phillip Pope, Yogesh Balaji, Soheil Feizi; Proceedings of the IEEE/CVF Conference on Computer Vision and Pattern Recognition (CVPR), 2022, pp. 19087-19097

**Questions:**

The term "historical attention fusion" needs more clarification.
Anyway, one needs to obviously learn/gather information features only from past frames, in any problem of video analytics.
So what is the reason of explicitly stating that using this term ? - unless there is significant meaning of something else -
say, state change in past in latent space, say.

Line 168  vs 182 - ...subscript "s" denotes parameters related to spatial features...
vs
M^st for temporal features, where "s' is a superscript.

the SUM function - limits/range of indices (even if obvious) not specified in Eqns. 1 & 2.

Eqns. (4-5)  -  sigmoid of a feature vector (as argument to the function) ? - something missing there. Pl. clarify

---

> ### Author Response · Authors · 2024-11-27
> **Provide additional explanations and rebuttals to the reviewers' comments**
>
> For Weaknesses：
>
> 1、State clearly the difference between the operations: historical attention fusion vs Spatio-temporal fusion.
>
> ==>answer:Thank you very much for your suggestion. This will help readers better understand the specific differences between the operations. I have added an explanation of the main differences between historical attention fusion and spatiotemporal fusion at the end of Section 3.3.2.
>
> 2、Difference between the terms information and feature (T & S) should be clarified - unless used interchangeably (specify so then).
>
> ==>answer:I have clearly provided the definition in the second paragraph of Section 3.3 of the article, and I have also provided consistent explanations for the areas that might cause confusion.
>
> 3、What about the use of any probabilistic model in prediction, future may be uncertain (although GT data may provide unique answer) ?
>
> ==>answer:I completely agree with your point. Indeed, in the real world, there is no single definitive answer. If we incorporate probabilistic models, it might better apply to real-life scenarios. This could definitely be a direction for further research in the future.
>
> 4、Fig. 1- text inside processing blocks barely legible
>
> ==>answer: Previously, in order to reduce the space occupied by the images, they were resized to the point where they became unreadable. I apologize for that. The images have now been enlarged.
>
> 5、Rightmost column of Fig. 7 - is mostly dark - not sure, what the authors want to show to readers.
>
> ==>answer: Because the image was resized too small, the differences, which were already subtle, were not very noticeable. Now that the image has been enlarged, the differences might be more apparent. To highlight the difference between the real frame and the predicted frame, you can see from the image that there are still clear contours depicting the differences. However, this might not appear as prominent when compared to images from different methods that are more visually intuitive.
>
> 6、Only for Kitti D/S the number of test frames is larger than training 10 -> 40, well not actually so, as I am pointing out to.
>
> ==>answer: I completely understand your point. The reason I handled and presented the final results this way is because I referred to the methods used in related papers, such as [1] CVPR 2023 'Temporal Attention Unit: Towards Efficient Spatiotemporal Predictive Learning' and [2] CVPR 2022 'SimVP: Simpler yet Better Video Prediction,' among a series of other papers that used the same approach.
>
> 7、The real challenge lies in predicting: (i) the number of predicted frames is large enough (longer duration), compared to history/past; (ii) HR frames from LR ones - did not find any such results(iii) is partly addressed, although better to show results on say: train: 10 --> 20 and then test 10--> 40 or 80/120 - what happens then.?
>
> ==>answer: Regarding the points you raised, you are absolutely right. How to improve the accuracy and stability of predictions over a longer time span is indeed an area that requires further research. This paper mainly focuses on the feature processing part. Thank you very much for your suggestion; the related experiments do indeed need to be addressed as well.
>
> 8、F/G-B/G separation is well-studied topic with vast literature, which is often difficult to cover in a Conf. paper. However, a few relevant papers which have not been cited, or performance not compared, are given below:
>
> ==>answer: Thank you very much for your sharing. Some of the papers I hadn't previously paid attention to, but after reading them, I gained a lot of valuable insights. I have also added citations for the papers I consider to be important. Thank you again.

---

> ### Author Response · Authors · 2024-11-27
> **Provide additional explanations and rebuttals to the reviewers' comments**
>
> For Questions：
>
> 1、The term "historical attention fusion" needs more clarification. Anyway, one needs to obviously learn/gather information features only from past frames, in any problem of video analytics. So what is the reason of explicitly stating that using this term ? - unless there is significant meaning of something else - say, state change in past in latent space, say.
>
> ==>answer: Historical attention fusion refers to the process of combining the current video frame with the temporal and spatial features of previous video frames. Based on their mutual correlations, attention scores are calculated separately to guide the model in learning the relevant key features from several previous frames of the video, using learned weights. Since the attention is computed based on the historical frames and the current frame, and the features are fused accordingly, this approach is named 'historical attention fusion.
>
> 2、Line 168 vs 182 - ...subscript "s" denotes parameters related to spatial features... vs M^st for temporal features, where "s' is a superscript.
>
> 2.1、the SUM function - limits/range of indices (even if obvious) not specified in Eqns. 1 & 2.
>
> ==>answer:As stated earlier in the formula, the ranges for
> 𝑖=1,…,𝜏 and 𝑗=1,…,𝜏 have already been clarified. I’m not sure which parameter’s index limit/range still needs to be specified. I hope you can provide a more detailed explanation.
>
> 2.2、Eqns. (4-5) - sigmoid of a feature vector (as argument to the function) ? - something missing there. Pl. clarify
>
> ==>answer:Eqns. (4-5) - sigmoid of a feature vector (as argument to the function)  is missing some details, and I couldn't quite understand it. Could you kindly explain it in more detail?

---

> > ### Comment · Reviewer_LpEN · 2024-12-02
> > **Feedback on Response/rebuttal by authors on review**
> >
> > Based on the response received from authors,
> > as well as observing the issues raised by other reviewers for this paper,
> > it appears that the majority opinion is not inclined towards acceptance.
> >
> > Most of my queries have been answered satisfactorily, but the overall method still has a lack of sufficient novelty
> > to get accepted for ICLR (my personal opinion). Hence, I am not changing my rating given earlier.
> >
> > Ofcourse, I leave the final decision to the ACs and PCs.

---

### Official Review · Reviewer_jgB3 · 2024-11-03

**Soundness:** 2
**Presentation:** 1
**Contribution:** 2
**Rating:** 3
**Confidence:** 3

**Summary:**

This paper proposes a video prediction model from the perspective of foreground and background separation. This practice may be novel but the motivation is not well related to the solution in this paper. The proposed solution is not novel in my view. The main reason for my rejection rating is the current version may be unfinished and needs major revisions.

**Strengths:**

1. Divide the video prediction goal into foreground and background parts and process them separately before aggregation.

**Weaknesses:**

1. The presentation of the entire paper needs to be carefully improved. For example:
a) The latex citet and citep commands should be used properly.
b) Fig 1 is too small.
c) Figures 3, 4 and 5 express essentially the same process and should be illustrated with a clearer diagram.
d) The visual comparisons are basically not distinguishable, especially in Figure 7 and 8.
e) Tables 1 to 4 are arranged in an order that does not match the content.
f) Writing and Grammar Issues.

2. The motivation and solution of the paper are not well related and do not clearly illustrate the differences with existing video prediction methods. From my point of view, the contribution is also insufficient and the proposed model components are not novel.

**Questions:**

1. Fonts differ from the official template.
2. In Table 2, we can find that compared to SimVPv2 the SSIM is better yet the PSNR is worse, which is an interesting observation that is worth analyzing.
3. Why the LPIPS metric is not used in the comparison experiments, but only in the ablation experiments.
4. Why is there no quantitative and qualitative comparison of training on the KITTI training set and testing on the KITTI test set? It has been done in existing methods including [1].

[1] A Dynamic Multi-Scale Voxel Flow Network for Video Prediction, CVPR 2023

---

> ### Author Response · Authors · 2024-11-27
> **Provide additional explanations and rebuttals to the reviewers' comments**
>
> For Weaknesses:
>
> The presentation of the entire paper needs to be carefully improved.
> a) The latex citet and citep commands should be used properly.
> ==>answer:Thank you very much for your guidance. I hadn't noticed these grammatical differences before, and I appreciate you pointing them out, as it has helped me learn something new.
>
> b) Fig 1 is too small. d) The visual comparisons are basically not distinguishable, especially in Figure 7 and 8.
>
> ==>answer:Because the journal has a page limit for the paper, I tried to reduce the size of the images as much as possible, which resulted in some formatting issues and the images not being very clear. I have made adjustments now, and I sincerely apologize for the inconvenience.
>
> c) Figures 3, 4 and 5 express essentially the same process and should be illustrated with a clearer diagram.
>
> ==>answer:I completely understand your point. I have also tried other types of charts, but in the end, I felt that this way of presentation might be easier for readers to understand. Other types of charts were harder to integrate together.
>
> e) Tables 1 to 4 are arranged in an order that does not match the content.
>
> ==>answer:In order to reduce the length of the paper, I made adjustments and consolidations to the table order, which may have caused the sequence to appear inconsistent. I have now made the necessary adjustments.
>
> f) Writing and Grammar Issues.
>
> ==>answer:I have already checked the relevant content, but could you please elaborate a bit more?
>
> 2、The motivation and solution of the paper are not well related and do not clearly illustrate the differences........
>
> ==>answer:I don’t fully agree with your point. When you say that the motivation and solution of the paper are not well connected, I explained the different characteristics of foreground and background features in video prediction, and based on this, I proposed a foreground-background separation solution for video prediction research and discussion. So, I'm not sure where you're coming from with that point. As I mentioned in the abstract, there are no similar ideas to mine in the current video prediction field, which is the most obvious distinction. I don't think it's necessary to explicitly list this out, especially since other papers on video prediction have also mentioned the different characteristics of their approaches.
>
> For Questions:
> 1、Fonts differ from the official template.
>
> ==>answer:I followed the official template to write the paper. To prevent any omissions, I downloaded the template again and rewrote it. If you notice any issues, please let me know, and I will make further revisions.
>
> 2、In Table 2, we can find that compared to SimVPv2 the SSIM is better yet the PSNR is worse, which is an interesting observation that is worth analyzing.
>
> ==>answer: Thank you for pointing this out. I compared my source code with the source code of SimVPv2 and found a difference in the way PSNR is calculated. The difference is in the way SimVPv2 computes the MSE:
> SimVPv2's MSE calculation:
>
> `mse1 = np.mean((np.uint8(pred * 255) - np.uint8(true * 255)) ** 2)`
>
> My method:
>
> `mse2 = np.square((pred * 255) - (true * 255)).mean()`
>
> The use of `np.uint8` causes data bias in the MSE, which leads to an overestimation of the PSNR.
>
> 3、Why the LPIPS metric is not used in the comparison experiments, but only in the ablation experiments.
>
> ==>answer:In the comparison experiment, I also performed a comparison using the LPIPS metric. However, due to space limitations and for better presentation, it was not included. In the ablation study, since there was still some space available, I added it. If necessary, I can include it all.
>
> 4、Why is there no quantitative and qualitative comparison of training on the KITTI training set and testing on the KITTI test set?
>
> ==>answer:The experimental setup of this paper involves training the proposed model on the KITTI dataset and testing it on the Caltech Pedestrian dataset. The training on the KITTI training set and testing on the KITTI test set were naturally performed, but in order to directly showcase the final prediction results, only the analysis and comparison on the Caltech Pedestrian dataset are presented. This approach was also referenced from related papers on video prediction.

---

> > ### Comment · Reviewer_jgB3 · 2024-11-29
> >
> > Thanks to the authors for their detailed responses, but I am not convinced by some of them. Overall, I think the updated version still fails to meet the ICLR's standard, both in terms of presentations and contributions. Therefore, I will keep my initial rating.
> >
> > Since neither the submitted version nor the revised version reaches the full 10 pages, I believe there will be a better way of presentation to avoid the meaningless repetition of Figs 3 to 5 and the too small figures and tables of the experimental results. The official template uses the Times New Roman font, but this submission does not. The authors can compare it with other submissions.
> >
> > The separation idea is straightforward and the authors do not explicitly state the novelty of the proposed fusion module compared to existing fusion strategies. This issue was also raised by other reviewers. In addition, the following related papers may be worth discussing.
> >
> > Generating Videos with Scene Dynamics, NeurIPS 2016
> >
> > High Fidelity Video Prediction with Large Stochastic Recurrent Neural Networks, NeurIPS 2019
> >
> > Hierarchical Long-term Video Prediction without Supervision, ICML 2018
> >
> > Self-Supervision by Prediction for Object Discovery in Videos, ICIP 2021
> >
> > Render In-between: Motion Guided Video Synthesis for Action Interpolation, BMVC 2021
> >
> > SLAMP: Stochastic Latent Appearance and Motion Prediction, ICCV 2021
> >
> > MOSO: Decomposing MOtion, Scene and Object for Video Prediction, CVPR 2023

---

### Official Review · Reviewer_2qhP · 2024-11-05

**Soundness:** 3
**Presentation:** 3
**Contribution:** 3
**Rating:** 8
**Confidence:** 4

**Summary:**

The authors propose a  foreground-background separation video prediction. The proposed method named FBSVP consists of modules. First, an historical attention fusion module employs an attention mechanism. Second, temporal and spatial features are fused through a spatio-temporal fusion module. Third,a  foreground-background fusion module allows the authors to estimate the final video frame.

**Strengths:**

1) The paper is well organized and relatively well written.
2) The proposed method FBSVP is detailed and reproducible.
3) Experiments are conducted on various datasets such as KITTI and Caltech Pedestrian showing the superiority of FBSVP compared to ten previous methods.

**Weaknesses:**

1) There are missing references about key surveys in the field of background/foreground separation for novices.

M. Cristani, et al., “Background Subtraction for Automated Multisensor Surveillance: A Comprehensive Review”, EURASIP Journal on Advances in Signal Processing, 24 pages, Volume 2010, 2010.

B. Garcia-Garcia, et al., "Background Subtraction in Real Applications: Challenges, Current Models and Future Directions",  Computer Science Review, Volume 35, February 2020.

2) All over the paper, the authors cite two times as in the sentence "Wang et al. Wang et al. (2017) argued...". Please only let the second citation.

**Questions:**

It would be interesting to test the proposed method on the CDnet 2014 dataset. Indeed, it is a large-scale dataset used in MOD and it would be interesting if the method can help in this context. If yes, the authors have just to mention it.

---

> ### Author Response · Authors · 2024-11-27
> **Provide additional explanations and rebuttals to the reviewers' comments**
>
> For Weaknesses:
>
> 1、There are missing references about key surveys in the field of background/foreground separation for novices.
>
> ==>answer:Thank you very much for your sharing. I had not previously paid attention to these two papers, but after reading them, I gained a lot of valuable insights. I have also added the references. Thank you again.
>
> 2、All over the paper, the authors cite two times as in the sentence "Wang et al. Wang et al. (2017) argued...". Please only let the second citation.
>
> ==>answer:Thank you very much for your suggestion. I have added the missing references to the paper, which has made it more complete. The issue of citing the same paper twice has also been corrected. Thank you again.
>
> For Questions:
>
> 1、It would be interesting to test the proposed method on the CDnet 2014 dataset. Indeed, it is a large-scale dataset used in MOD and it would be interesting if the method can help in this context. If yes, the authors have just to mention it.
>
> ==>answer: Thank you very much for your suggestion. The CDnet 2014 dataset is very helpful for further testing the performance stability and robustness of the proposed method. I am currently conducting the tests and hope to have results soon that I can share with you

---

### Meta-Review · Area_Chair_mZxr · 2024-12-21

**Metareview:**

## Summary
The proposed training method for video prediction involves separating foreground and background, but has limitations due to the assumption that backgrounds change infrequently. Complex scenes require significant changes in both, making it challenging to perform video prediction on separately processed foreground and background.

## Strength
* Proposed method FBSVP is detailed and reproducible on experiments conducted on KITTI and Caltech Pedestrian.
* Video prediction goal divided into foreground and background parts for separate processing.
* Performance improvement achieved on multiple datasets.

## Weaknesses
* The paper's presentation needs improvement, including proper use of latex citet and citep commands, larger Fig 1, clearer diagrams in Figures 3, 4, and 5, and clearer visual comparisons.
* The motivation and solution of the paper are not well related and do not clearly illustrate differences with existing video prediction methods.
* The authors do not explicitly state the novelty of the proposed fusion module compared to existing fusion strategies.
* Video prediction can be considered a subtask of mainstream video generation, where the goal is to condition on the first few frames of a video to generate future content and compared are all based on supervised video prediction, lacking comparisons with mainstream video generation methods.

## Conclusions
Based on the reviews and the author's feedback, the paper should include all the missing references and also improve the text and the motivation of the paper before accepting the paper.

**Additional Comments On Reviewer Discussion:**

The reviewers stated several weak points and there were a lot of missing references that should be included before accepting the paper. Moreover, a few reviewers were satisfied with the feedback but not with the overall results.

---

### Decision · Program_Chairs · 2025-01-22

Reject